# Hierarchical nucleation in deep neural networks

**Diego Doimo**
International School for Advanced Studies
ddoimo@sissa.it

**Aldo Glielmo**
International School for Advanced Studies
aglielmo@sissa.it

**Alessio Ansuini**
Area Science Park
alessio.ansuini@areasciencepark.it

**Alessandro Laio**
International School for Advanced Studies
laio@sissa.it

## Abstract

Deep convolutional networks (DCNs) learn meaningful representations where data that share the same abstract characteristics are positioned closer and closer. Understanding these representations and how they are generated is of unquestioned practical and theoretical interest. In this work we study the evolution of the probability density of the ImageNet dataset across the hidden layers in some state-of-the-art DCNs. We find that the initial layers generate a unimodal probability density getting rid of any structure irrelevant for classification. In subsequent layers density peaks arise in a hierarchical fashion that mirrors the semantic hierarchy of the concepts. Density peaks corresponding to single categories appear only close to the output and via a very sharp transition which resembles the nucleation process of a heterogeneous liquid. This process leaves a footprint in the probability density of the output layer where the topography of the peaks allows reconstructing the semantic relationships of the categories.

## 1   Introduction

Deep convolutional networks (DCNs) have become fundamental tools of modern science and technology. They provide a powerful approach to supervised classification, allowing the automatic extraction of meaningful features from data. The capability of DCNs to discover representations without human input, has attracted the interest of the machine learning community. In the intermediate layers of a DCN, the data are represented with a set of features (the activations) embedded in a manifold whose tangent directions capture the relevant factors of variation of the inputs [1, 2]. Accordingly, understanding these data representations requires both studying the geometrical properties of the underlying manifolds and characterising the data distributions on them.

In the present paper, we analyse how the probability density of the data changes across the layers of a DCN. We consider in particular DCNs trained for classifying ImageNet; as we will see, the complexity and heterogeneity of this dataset critically affects the results of our analysis.

Comparisons between representations based on generalizations of multivariate correlation have already been performed with the methods in [3] (SVCCA), [4] (PWCCA) and, more extensively, in [5] (CKA). Representational similarity analysis (RSA) [6] – introduced originally in neuroscience – investigates artificial representations as well, and in each layer a matrix of pairwise distances (representation dissimilarity matrix (RDM) ) between the activation vectors of the data points tells which data is similar or dissimilar in that layer. The introduction of RDMs allowed performing multiple comparisons including those between artificial and biological networks [7, 8, 9].

More recently many techniques have been proposed to understand representations. In [10] it was shown that feature maps are often aligned to a specific explanatory factor of the data distribution

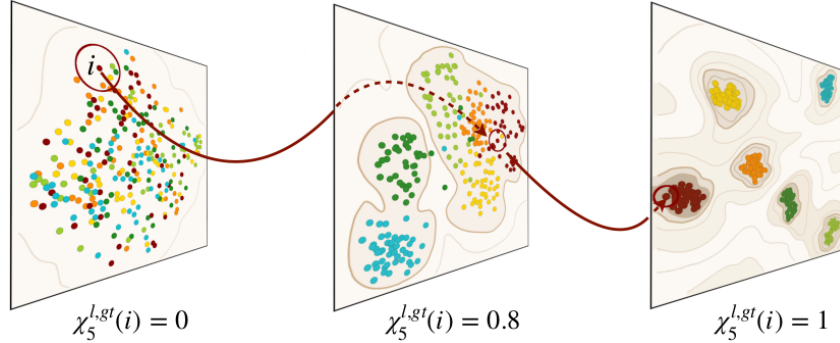

$$\chi_5^{l,gt}(i) = 0 \qquad \chi_5^{l,gt}(i) = 0.8 \qquad \chi_5^{l,gt}(i) = 1$$

Figure 1: **Evolution of the data representations in ResNet152**. Projections of the representations of the input (left), conv4 (middle) and output layers in ResNet152 for six ImageNet classes. Contours schematically portray the density isolines on the data manifold. The dark red circles surround the five nearest neighbors of a point $i$; $\chi_5^{l,gt}(i)$ represents the fraction of these points that are in the same class of $i$.

and therefore behave as detectors of specific concepts. In [11] linear separability of the features was shown to increase with a smooth monotonic trend across the internal layers. In [12] the question whether DCNs learn a hierarchy of classes was addressed exploiting class confusion patterns. Other studies investigated more specifically geometrical and structural properties of the representations. In [13] a common trend in the intrinsic dimension was found across several architectures; in [14] the soft-neighbor loss was used as a tool to reveal structural changes in neighbors organization across layers, also during training [15].

Here we take a complementary perspective. One can view a DCN as an engine capable of iteratively shaping a probability density. Input data can be seen as instances harvested from a given probability distribution. This distribution is then modified again and again by applying, at each layer, a non-linear transformation to the data coordinates. The result of this sequence of transformations is well understood: in the output layer of a trained network, data belonging to different categories form well separated clusters, which can be viewed as distinct peaks of a probability density (see Fig. 1). But where in the network do these peaks appear? Do they develop slowly and gradually or all of a sudden? Is this change model-specific or is it shared across architectures? And what is the probability *flux* between a layer and the next? In the input layer the data points are mixed: data with different ground truth labels are close to each other. In the output layer, the neighborhood of each data point is ordered, namely it contains mostly data points belonging to the same category. Where in the network does the transition from disordered to ordered neighborhoods take place? Is it simultaneous with the formation of the probability peaks?

The pivotal role of depth in determining the accuracy of a neural network suggests that the transformation of the probability density should be slow and gradual in order to be effective. The analysis reported in [3, 4, 5] are consistent with this scenario. However, we will see that especially in a DCN trained for a complex classification task the evolution of the probability density is not really smooth, with spikes in the probability flux and sudden changes in the modality. We analyse the probability landscape in the intermediate layers of a DCN by a technique which estimates the probability density and characterizes its features even if this is defined as a function of hundreds of thousands of coordinates, provided that the data are embedded in a relatively low dimensional manifold [16, 17]. This approach *does not* require a dimensional reduction of the data and has the great advantage that the embedding manifold does not necessarily have to be an hyperplane, but can be arbitrarily curved, twisted and topologically complex. To analyse the probability flux between the layers we use an extension of neighboring hit [18]. The main results of this analysis also sketched in Fig. 1 can be summarized as follows:

- Representations in DCNs trained for complex classification tasks do not evolve smoothly, but through nucleation-like events, in which the neighborhood of a data point changes rather suddenly (Sec. 3.1);

- In the first layers of the network any structure which is initially present in the probability density of the input is washed out, reaching a state with a single probability peak where

the neighborhoods mainly contain simple images characterized by elementary geometrical shapes (Sec. 3.2);

- In the successive layers a structure in the landscape starts to emerge, with probability peaks appearing in an order that mirrors the semantic hierarchy of the dataset: neighborhoods are first populated by images that share the same high-level attributes (Sec. 3.3);

- In the output layer the probability landscape is formed by density peaks containing data points with the same ground truth classification; interestingly, these peaks are organized in complex "mountain chains" resembling the semantic kinship of the categories (Sec. 3.3).

It short, we find that the disorder-order transition induced by a trained DCN can be characterized without any reference to the ground truth categories as a sequence of changes in the modality of the probability density of the representations. These changes are achieved by reshuffling the neighbors of the data points again and again, in a process which resembles the diffusion in an heterogeneous liquid, followed by the nucleation of an ordered phase.

## 2  Methods

The DCNs we consider in this work are classifiers $\mathbf{y} = f(\mathbf{x})$ that map a data point $\mathbf{x}_i \in \mathbb{R}^p$, for example an image, to its categorical target $\mathbf{y}_i \in \{0,1\}^q$ typically encoded with a one-hot vector of dimension $q$ equal to the number of classes. Feedforward networks achieve the task via a function composition $f = f^{(1)} \to f^{(2)} \to ... \to f^{(L)}$ that transform the input sequentially $\mathbf{x}_i \to \mathbf{x}_i^{(1)} \to ... \to \mathbf{x}_i^{(L)}$. We call the vector $\mathbf{x}_i^{(l)}$ containing the value of the activations of the $l$-th layer for data point $i$ the *representation* of $\mathbf{x}_i$ at the layer $l$. The sequence of representations of these datapoints on a trained network can be seen as a "trajectory" in a very high dimensional space. The relative positions of the $N$ inputs change from an initial state where the neighborhood of each point contains members of different classes to a final state where images of the same class have been mapped close together to the same target point. We study this process with two approaches, one aimed at describing the probability flux across the layers (Sec. 2.1) the other aimed at characterizing the features of the probability density in each layer (Sec. 2.2).

### 2.1  The neighborhood overlap

Let $\mathcal{N}_k^l(i)$ be the set of $k$ points nearest to $\mathbf{x}_i^{(l)}$ in euclidean distance at a given layer $l$, and let $A^l$ be an $N \times N$ adjacency matrix with entries $A_{ij}^l = 1$ if $j \in \mathcal{N}_k^l(i)$ and 0 otherwise. Through $A$ we define an index of similarity $\chi_k^{l,m} \in [0,1]$ between two layers $l$ and $m$ as:

$$\chi_k^{l,m} = \frac{1}{N} \sum_i \frac{1}{k} \sum_j A^l{}_{ij} A_{ij}^m \tag{1}$$

The similarity just introduced has a very intuitive interpretation: it is the average fraction of common neighbors in the two layers considered: for this reason, we will refer to $\chi_k^{l,m}$ as the *neighborhood overlap* between layers $l$ and $m$.

In the same framework we also compare the similarity of a layer with the ground truth categorical classification defining the "ground truth" adjacency matrix $A_{ij}^{gt} = 1$ if $y_i = y_j$ and 0 otherwise. In this case $\chi_k^{l,gt} = \frac{1}{N} \sum_i \frac{1}{k} \sum_j A^l{}_{ij} A_{ij}^{gt}$ is the average fraction of neighbors of a given point in $l$ that are in the same class as the central point (see Fig. 1). We set $k$ to one tenth of the number of images per class, but we verified that our findings are robust with respect to the choice of $k$ over a wide range of values (see Sec. A.1). $A^l{}_{ij}$ and $A_{ij}^{gt}$ are built using the euclidean distances between images, as such they are invariant to orthogonal transformations but not to any arbitrary linear transformation of the activations. These convenient properties for similarity indices between representations [5] are inherited by $\chi_k^{l,m}$ and $\chi_k^{l,gt}$. When calculated using the ground truth adjacency matrix as a reference, $\chi_k^{l,gt}$ reduces to the neighboring hit [18]. A measure of overlap quantitatively similar to $\chi_k^{l,m}$ can be obtained by using the CKA method [5] with a gaussian kernel of very small width (see Sec. A.4).

## 2.2 Estimating the probability density

We analyse the structure of the probability density of data representations following the approach in [16, 17], which allows to find the peaks of the data probability distribution and the location and the height of the saddle points between them. This in turn provides information on the relative hierarchical arrangement of the probability peaks.

The methodology works as follows. Using a kNN estimator the local volume density $\rho_i$ around each point $i$ is estimated. The maxima of $\rho_i$ (namely the probability peaks) are then found. Data point $i$ is a maximum if the following two properties hold: (I) $\rho_i > \rho_j$ for all the points $j$ belonging to $\mathcal{N}_k(i)$; (II) $i$ does not belong to the neighborhood $\mathcal{N}_k(j)$ of any other point of higher density [16]. A different integer label $\mathcal{C} = \{c^1, ... c^n\}$ is assigned to each of the $n$ maxima, and the data points that are not maxima are iteratively linked to one of these labels, by assigning to each point the same label of its nearest neighbor of higher density. The set of points with the same label corresponds to a *probability peak*.

The saddle points between probability peaks are then found. A point $\mathbf{x}_i \in c^\alpha$ is assumed to belong to the border $\partial_{c^\alpha, c^\beta}$ with a different peak $c^\beta$ if it exists a point $\mathbf{x}_j \in \mathcal{N}_k(i) \cap c^\beta$ whose distance from $i$ is smaller than the distance from any other point belonging to $c^\alpha$. The saddle point between $c^\alpha$ and $c^\beta$ is the point of maximum density in $\partial_{c^\alpha, c^\beta}$.

Finally, the statistical reliability of the peaks is assessed as follows. Let $\rho^\alpha$ be the maximum density of peak $\alpha$, and $\rho^{\alpha, \beta}$ the density of the saddle point between $\alpha$ and $\beta$. If $\log \rho^\alpha - \log \rho^{\alpha, \beta} < 2Z\sqrt{(4k+2)/[k(k+1)]}$, peak $\alpha$ is merged with peak $\beta$ since the value of its density peak is considered indistinguishable from the saddle point at a statistical confidence fixed by the parameter $Z$ [16]. The process is repeated until all the peaks satisfy this criterion, and are therefore statistically robust with a confidence $Z$.

## 2.3 The dataset and the network architecture

We perform our analysis on the ILSVRC2012 dataset, a subset of 1000 mutually exclusive classes of ImageNet which can be considered leaves of a hierarchical structure with 860 internal nodes. The highest level of the hierarchy contains seven classes but $95\%$ of the ILSVRC2012 images belong to only two of these (artifacts or animals) and are split almost evenly between them ($55\%$ and $45\%$ respectively). Unless otherwise stated, the analysis in this work is performed on a subset of 300 randomly chosen categories, including 300 images for each category, for a total of $90,000$ images.

We extracted the activations of the training set of ILSVRC2012 from a selection of pretrained PyTorch models: ResNets [19], DenseNets [20], VGGs [21] and GoogleNet [22]. To compare architectures of different depths we will use as *checkpoints* the layers that downsample the channels and the final fully connected layers. In these layers the learned representations become more abstract and invariant to details of the input irrelevant for the classification task [1].

**Reproducibility**  The source code of our experiments with the instructions required to run it on a selection layers is available at https://github.com/diegodoimo/hierarchical_nucleation as well as in the online supplementary material.

## 3 Results

It is well known that neural networks modify the representations of the data from an initial state where all the data are randomly mixed, to a final state where they are orderly clustered according to their ground truth labels [15]. But where in the network does this order arise? In the output layer the nearest neighbors of, say, the image of a cat are very likely other images of cats. But in which layer, and in which manner do cat-like images come together? We describe the ordering process by analyzing the change in the probability distribution across the layers.

### 3.1 The evolution of the neighbor composition in a DCN

We first characterize the probability flux by computing the neighborhood overlap $\chi_{30}^{l, out}$: the fraction of 30-neighbors of a data point which are the same in layer $l$ and in the output layer (Eq. 1). For the

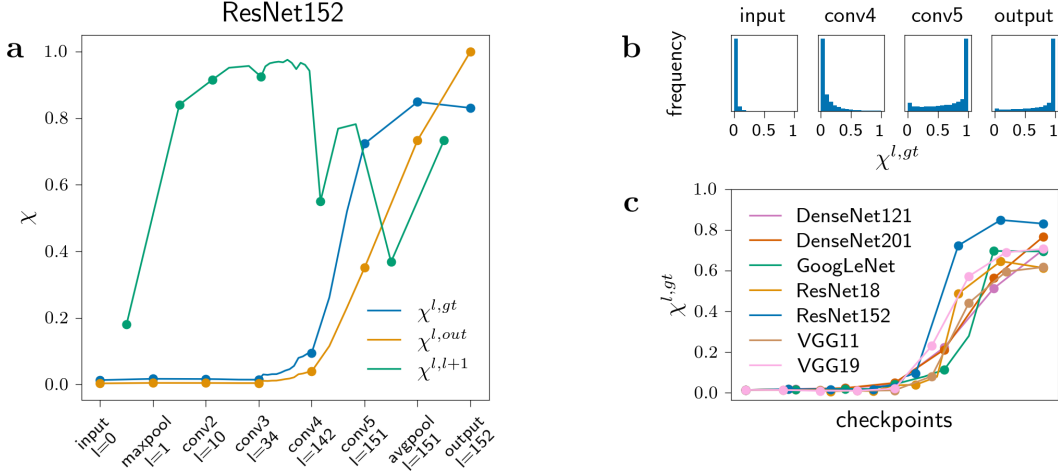

Figure 2: **Overlap profiles in ResNet152 and in different architectures**. (**a**:) Overlap with the ground truth $\chi^{l,gt}$ (blue), with the output $\chi^{l,out}$ (orange) and between nearby layers $\chi^{l,l+1}$ (green). Checkpoint layers are evenly spaced on the $x$-axis and marked with dots. (**b**:) Probability distribution of $\chi^{l,gt}$ approximated by its frequency histogram. The histograms are built collecting $\chi^{l,gt}(i) = \sum_j A^l_{ij} A^m_{ij}/k$ (see Eq. 1) for each data point $i$ in four layers. (**c**:) Profiles of $\chi^{l,gt}$ for seven architectures of different depths. The values of $\chi^{l,gt}$ measured on the checkpoints are evenly spaced on the $x$-axis.

sake of brevity for now on we will omit the subscript $k = 30$. Figure 2-a shows the behaviour of $\chi^{l,out}$ as a function of $l$ for the checkpoint layers of the ResNet152 described in Sec. 2.3 (orange line). The neighborhood overlap remains close to zero up to $l$=142. In the next nine layers it starts growing significantly, reaching a value of 0.35 in layer 151 and 0.73 in layer 152, the last before the output. In the same figure, we also plot the neighborhood overlap of each layer with the ground truth classification $\chi^{l,gt}$ (blue line). After layer 142, $\chi^{l,gt}$ changes even more abruptly than $\chi^{l,out}$, increasing from 0.10 to 0.72.

We can obtain more insights into this transition process looking at the probability distribution of $\chi^{l,gt}$ across the dataset in four different layers (see Fig. 2-b). In the input and output layers the probability distribution is unimodal. In layer 142 (conv4), before the onset of the transition, the distribution is still strongly dominated by disordered neighbors (i.e. $\chi^{l,gt} \approx 0$ for most of the data points), but an ordered tail starts to emerge. In layer 151 (conv5), immediately after the transition, the distribution indicates the coexistence of some data points in which the neighborhood is still disordered or only partially ordered ($\chi^{l,gt} \approx 0$) and some data in which it is already ordered ($\chi^{l,gt} \approx 1$).

These results show that ordering, when measured by the consistency of the neighborhood of data points with respect to their class labels, changes abruptly, in a manner which qualitatively resembles the phase transition of a "nucleation" process. The green profile of 2-a reinforces this evidence showing the overlap between two nearby layers $\chi^{l,l+1}$. This quantity is a measure of the probability flux between any two consecutive layers. In the first layer the neighborhoods are almost completely reshuffled, as indicated by an $\chi^{0,1} \sim 0.2$. Afterwards, up to layer 142 $\chi^{l,l+1} \sim 0.95$, indicating that the neighborhoods change their compositions smoothly like in a slow diffusion process. In the two central blocks, from layer 10 to layer 142, it takes 20-30 layers to change half of the neighbors of each data point, i.e. to decrease $\chi^{l,l'}$ to 0.5 (see Sec. A.3). At layer 142 instead, the first ordered nuclei appear and $\chi^{l,l'}$ drops to 0.55 in just one layer. A significant reshuffling of the neighborhoods takes place at layer 151, where $\chi^{l,l+1}$ drops again to 0.61. We will see in Sec. 3.3 that in this layer the structure of the probability density changes significantly, and the probability peaks corresponding to the "correct" categories appear. The effective "attractive force" acting between data with the same ground truth label overcomes the entropic-like component coming from the intrinsic complexity of the images, and clusters of akin images emerge almost all at the same time (i.e., at the same layer), giving rise to a sharp transition.

Is the sudden change we observed specific to this architecture or is it a common feature of deep networks trained for similar tasks? To answer this question we repeated the same experiment on other ten networks including ResNets, DenseNets, VGGs and GoogleNet. Because of space limitations we

report the results of $\chi^{l,gt}$ on seven architectures (see Fig. 2-c). The remaining tests can be found in A.2.

Fig. 2-c reveals a common trend in all the architectures: $\chi^{l,gt}$ remains close to zero for many layers and then sharply increases in only a few layers towards the end of the networks. The value of $\chi^{l,gt}$ in the output layer is different in different architectures, consistently with the fact that their classification accuracy is different.

## 3.2 The data landscape before the onset of ordering

It has been argued that the first layers of deep networks serve the important task of getting rid of unimportant structures present in the dataset [23, 24, 13, 25]. This phenomenon is illustrated in the upper panel of Fig. 3, which shows that any overlap with the input layer is lost roughly after the conv3 landmark (layer 34).

In the intermediate layers the DCNs analysed in this work construct high-dimensional hyperspherical arrangements of points with very few images at the center. This is related to the high intrinsic dimension (ID) of these layers [13]. When the ID is very high, few data points act as "hubs"[26], namely they fall in a large fraction of the other point's neighborhoods while the others fall in just a few.

Moving from the input to conv3 the same images appear in a growing number of neighborhoods. In conv3 the top ten most frequent images are found in almost half of the 90,000 neighborhoods with a high of 75,663 for the most frequent of all.

Interesingly, we found that hub images are particularly "simple", looking in most cases like elementary patterns (dots, blobs, etc.) lying on almost uniform backgrounds (see Fig. 3, bottom right). To quantify this perceptual judgment we computed the Shannon entropy of an image $S = -\sum_{n_c}\sum_v p_v \log_2(p_v)/n_c$ where $p_v$ is the normalized frequency of pixels of value $v$ and $n_c = 3$ is the number of channels of RGB images. The average entropy of the neighbors of an image $i$ in a layer $l$ is given by $S_i^l = \sum_{j \in \mathcal{N}_k(i)} S_j^l/k$, and averaging across all images we obtain a measure of the neighborhood entropy of a layer $S^l$. A low value of $S^l$ means that, in that layer, the neighborhoods contain many low-entropy images. In bottom panel of Fig. 3 we show how $S^l$ changes: in intermediate layers, and most prominently in conv3 – where the intrinsic dimension is at its peak (see Sec. A.5) – the representation is organized around low-entropy hubs whose centers are low-$S$ images (blue line, and left stack of hub images). As a reference, we also report the entropy computed shuffling the neighbors assignments (grey dashed line).

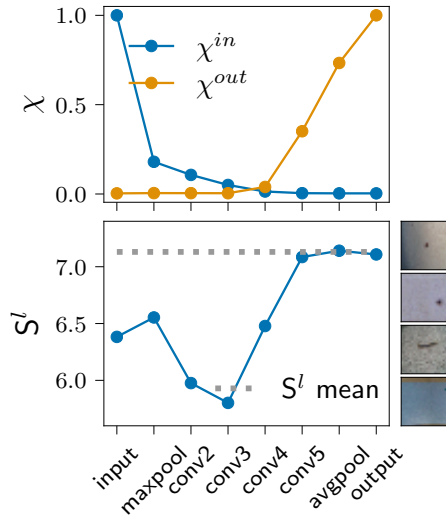

Figure 3: **Image entropy in ResNet152.** (**Top**:) Overlap with the input $\chi^{in}$ (blue), and output $\chi^{l,out}$ (orange) layers. (**Bottom**:) Average image image entropy $S^l$ within the first 30 neighbors; the errorbars are shorter than the marker size; the most frequent images found in conv3 are shown on the right.

## 3.3 The evolution of the probability density across the hidden layers

We have seen that at layer34 (conv3) all images are arranged in a high-dimensional hypersphere and that at layer 151 (conv5) the neighborhoods are already organized consistently with the classification. Clearly, the most important transformations of the representation happen in between these layers. To shed some light on the evolution of the representations in these intermediate layers we use a tool which characterizes a multidimensional probability distribution, finding its probability peaks and localizing all the saddle points separating the peaks. Unlike other methods for the analysis of hidden representations based on dimensional reduction [27, 28] here we do not perform any low dimensional embedding of the data. In this respect we stress that Fig. 4a is used only to aid the visualization of the peaks of the multidimensional probability distribution, which are obtained with the algorithm in Sec. 2.2. We will see that the nucleation-like transition described in Sec. 3.1 is a complex process, in which the network separates the data in a gradually increasing number of density

peaks laid out in a hierarchical fashion that closely mirrors the hyperonymous-hyponymous relations of the ILSVRC2012 dataset.

Figure 4-a shows a two-dimensional visualization of the number and organization of the probability peaks of the representations in some of the layers. In the input layer ($l=0$), the data are split into two major peaks, which roughly divide the training set into light and dark images. This structure is not useful for classification, and is wiped out within the first 34 layers of the network. In conv3 the probability density becomes unimodal, consistently with the analysis of the previous section. In the subsequent layers the network creates structure that is useful for classification, and in layer 97 a bimodal distribution appears. The other peaks shown in figure are very small and retain only a few hundreds data points each. The same density peaks persist until layer 142, where 97% of the images still reside in the two biggest ones. Finally, after layer 142 the two large peaks break down into smaller ones representing individual classes.

To asses the population of the density peaks in terms of ground-truth categories we use the Adjusted Rand Index ($ARI$) [29, 30]. Roughly speaking, $ARI$ is zero if the density peaks do not correspond to the reference partitions of the dataset, and is one if they match it. In Fig. 4-b we plot the $ARI$ with respect to the high level animal/artifact categories ($ARI^{macro}$, orange line), and with respect to the 300 low level classes we sampled ($ARI^{cl}$, blu line). From layer 97 to layer 142 artifacts and animals predominantly populate one of the two major peaks increasing the $ARI^{macro}$ value to 0.22 while the correlation with low level classes remains absent. The following breakup of the peaks leads to a drop of $ARI^{macro}$ to 0 and to a concomitant sharp rise of $ARI^{cl}$ from 0 to 0.55, consistent with

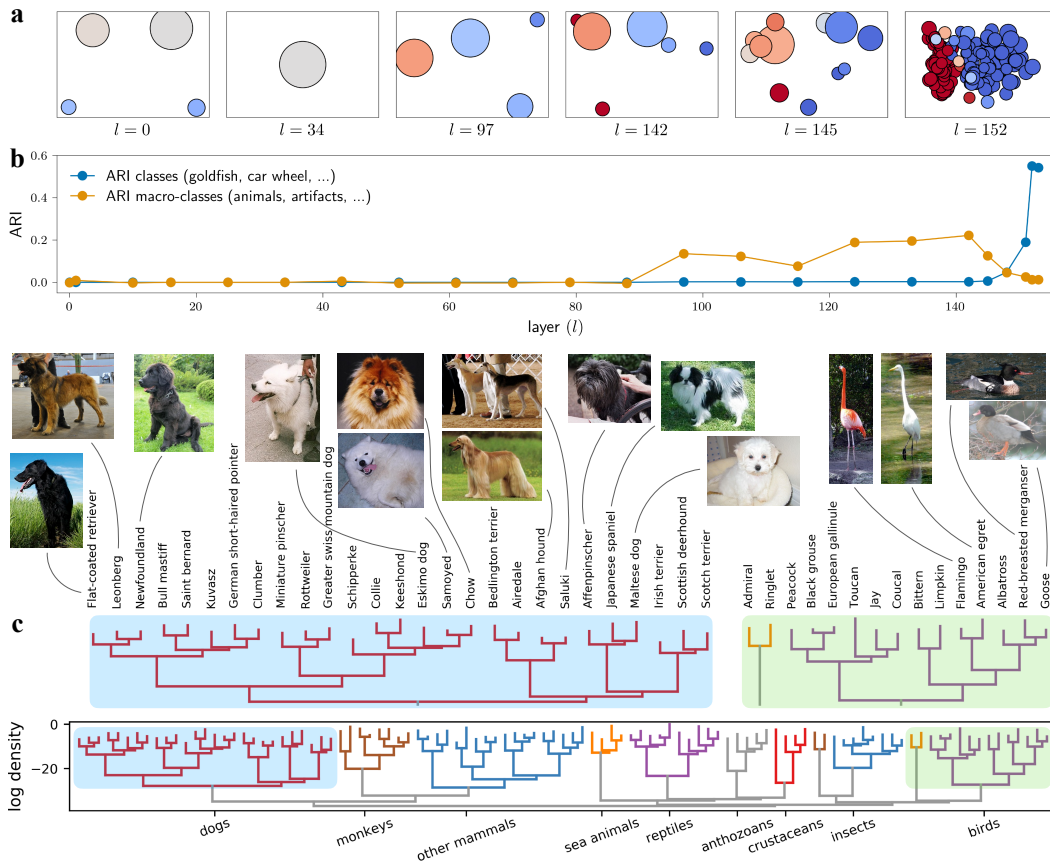

Figure 4: **Structure and composition of density peaks of representations**. (**a**:) A schematic view of the peaks in 6 layers. Color tones refer to the relative presence of animals and artifacts in each peak: dark red = 100% of animals, dark blue = 100% of artifacts. (**b**:) $ARI$ profiles for animal/artifact partition and the 300 low-level classes (blue and orange). (**c**:) The dendrogram portrays the hierarchical connections between the density peaks of the animal branch. On the $y$-axis the value of the density peaks is plotted in logarithmic scale. Two insets above show the detailed composition of specific branches (light blue and light green).

the nucleation mechanism detected by $\chi^{l,gt}$ and described in Sec. 3.1. Moreover, some classes are separated before others (Fig. 4-a, layer 145), consistently with the bimodality in the distribution of $\chi^{l,gt}$ observed in the bottom panels of Fig. 2-a. Interestingly, many of the density peaks in the layers between 142 and 153 (i.e., during the nucleation transition) closely resemble the hierarchical structure of the concepts in ILSVRC2012. For instance, in layer 148 one can find peaks corresponding to insects, birds, but also ships and buildings (see Sec. A.6).

In the last layer ($l = 153$) the different peaks correspond to the different classes, but the structure of the probability density is much richer than a simple collection of disjointed peaks. Indeed the hierarchical process that shaped the density landscape across the layers leaves a footprint on the organization of the peaks. For instance, the division in macro-classes of animals and artifacts formed in layer 97 is still present in the last layer as indicated by the fact that red and blue clusters are found primarily on the left and on the right of the corresponding low dimensional embedding (Fig. 4-a). But much more structure is present. In Figure 4-c we visualize the probability landscape of the animal classes as a dendrogram, in which each leaf corresponds to a peak, and the leaves are merged sequentially, following the WPGMA algorithm [31], according to the height of the saddle point of the probability density between them. In this manner the secondary probability peaks belonging to the same large scale structure form a branch of the dendrogram. The height of a leaf in Fig. 4-c is proportional to the logarithm of the density of the peaks. The morphological similarities of animals with similar genetic material make it possible for the dendrogram in Fig. 4 to reproduce the taxonomy of a phylogenetic tree to an astonishing degree. At the root of the dendrogram, we can notice a first distinction between mammals on the left and other animals on the right. At the following herarchical level we can find a more specific separation of animal types. Dogs, reptiles, birds and insects and so on can be easily identified. Finally, within each species, say dogs (Fig. 4), alike breeds are linked by tighter bounds, that is saddle points of higher density.

In the supplementary material we include the values of the probability density and the integer identifier of the density peak to which each image belongs for the relevant representations analysed in this section. We also provide the topography of the probability density, namely the height of all the peaks and of all the saddle points between them.

## 4  Discussion

This paper presents an explicit characterization of the evolution of the probability density of the data landscape across the layers of a DCN. We showed that this probability density undergoes a sequence of transformations which brings to the emergence of a rugged and complex probability landscape. Rather surprisingly, we found that the development of these structures is not gradual, as one would expect in a deep network with more than one hundred layers. Instead, the greatest changes to the neighborhood composition and the emergence of the probability peaks are localized in a few layers. This picture seems qualitatively different with the one emerging from SVCCA [3], projection weighted CCA [4] and linear CKA [5], which have revealed smoother changes between nearby representations. A first reason of this difference lies in the kind of correlation captured by these similarity indices. The ordering mechanism starting with the separation between animals and artifacts is functional and correlated to a successful fine-grained classification of the categories. In essence CCA based methods capture the correlation between the final categories (the peaks) and the "intermediate level" concepts ("the mountain chains") required to construct them which are recognized already in the middle layers of the network. The overlap defined in Eq. 1 measures instead a correlation growing only when, locally, the neighbors become consistent with those of the output ($\chi^{l,out}$) or their labels ($\chi^{l,gt}$). In section A.4 of the appendix we show that $\chi^{l,out}$ is similar to the correlation obtained by Gaussian CKA[5] using a very small kernel width.

A second possible reason for the discrepancy between the results reported in this work and those reported in the literature is the complexity of the datasets analysed. Indeed, most previous studies have focused on datasets like MNIST and CIFAR-10. These datasets lack the semantic stratification of ImageNet and hence show a much smoother evolution of the probability landscape, because in these datasets the number steps needed to disentangle the hierarchy of features of the categories is smaller (see Sec. A.5). We are unaware of attempts that directly targeted the similarity of DCN representations in connection to the hierarchical structure of a complex dataset like ImageNet. In [12], confusion matrices were used to visually analyse the correlations between classes showing results in agreement with our conclusions. However, the algorithm we use here (Sec. 2.2) relies on density

estimation and is able to reconstruct a probability landscape that faithfully follows the hierarchical structure of categories (Sec. 3.3) in an unsupervised manner, with no need to consider the ground truth labels and estimating the confusion matrix; indeed, our approach works also in the limiting case of 100% test accuracy.

## Broader impact

We believe that the detailed picture of the evolution of the probability density provided in this work can improve the performance of learning protocols, make transfer learning more effective and enrich the information extracted from a multiclass classifier.

For example, the knowledge of the probability landscape can provide a rational criterion to improve training schemes based on triplet loss [32]. In this setting it is crucial to select challenging triplets where the so-called "anchor" image is closer to an example of a different class (*negative*) than to that of the same class (*positive*). Datapoints lying at the boundary of a probability peak could be used as anchors as they are on average closer to negatives than those lying close to cluster centers. One can also imagine to define training losses targeting the development of probability peaks according to a pre-defined semantic classification. This can be enforced in the intermediate layers of a network going well beyond a simple disentanglement of the feature space [33], enhancing the separation between macro categories which arise spontaneously.

An appropriate understanding of the nucleation mechanism could also be beneficial to transfer learning, since it gives a simple rational criterion to judge the generality of the features of a representation [34].

Finally, one can imagine to use the topography of the density peaks developed by a deep neural network as a hierarchical classifier, going beyond the sharp classification in mutually exclusive categories [35].

## Acknowledgements

The authors would like to thank Federico Barone, Davide Zoccolan, Alex Rodriguez and Jakob Macke for useful discussion and many precious suggestions. This work was supported by the International School for Advanced Studies (SISSA).

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
