[Supplementary Material]

# A Appendix

## A.1 Scaling of $\chi^{l,gt}$ and $\chi^{l,out}$

Figure A.1-a shows the overlap with the ground truth $\chi^{l,gt}$ (top) and with the output activations $\chi^{l,out}$ (bottom) in ResNet152, for the same subset of 90,000 examples from ILSVRC2012 analysed in the paper. In our experiments we empirically set $k = 30$ i.e. one tenth of the number of images per class. Figure A.1-a shows that the trend of $\chi^{l,gt}$ and $\chi^{l,out}$ is rather robust over a wide range of $k$-values. Only when $k$ is very large ($k = 300$) the transition in the last layers of the network is not detected very clearly.

In Figure A.1-b we plot how $\chi^{l,gt}$ (top) and $\chi^{l,out}$ (bottom) vary with the dataset size $N$. As the number of examples $N$ increases we keep the ratio between the number of classes and images per class constant. This shows that the results are also robust with respect to $N$.

Figure A.1: (**a**:) Profiles of the overlap with the ground truth labels (top) and with the output layer (bottom) as a function of the neighbor size. (**b**:) Profiles of the overlap with the ground truth labels (top) and with the output layer (bottom) as a function of the total number $N$ of images.

## A.2 Overlaps vs relative depth

Figure A.2 shows the profiles of $\chi^{l,gt}$ (top panels) and $\chi^{l,out}$ (bottom panels) for ten deep neural networks. The number of layers of each architecture can be inferred from the name with the sole exception of GoogLeNet which has 22 layers.

All the profiles increase significantly only close to the output. The left panels also show that both $\chi^{l,gt}$ and $\chi^{l,out}$ tend to increase more abruptly and closer to the output layer the deeper is the network. The profiles show a monotonic increasing trend with the exception of the DensNet layers in a dense block [1]. Within a dense block all the layers are connected with subsequent ones. Feature maps are therefore markovian only between the transition layers at the end of the blocks. In the main text we have considered only markovian feature maps: in this case $\chi^{l,gt}$ and $\chi^{l,out}$ are always monotonic increasing functions of the layer depth.

Figure A.2: Profiles of $\chi^{l,gt}$ (top panels) and $\chi^{l,out}$ (bottom panels) for a subset of layers of ten deep neural networks. The layers that downsample the channels (checkpoints) are represented with dots. On the $x$-axis of the left panels we rescale the layer depths by the network size, on the $x$-axis of the right panels we instead display the checkpoints uniformly spaced.

## A.3 Overlap with the checkpoint layers

Figure A.3 shows the overlap of the representations with respect to the representation at tree layers $l = 25$, $l = 88$ and $l = 148$, belonging to tree distinct ResNet152 blocks. On average the number of layers required to change half of the neighbors is $\sim 20$ in conv3 and $\sim 30$ in conv4, while in conv5 where the nucleation takes place the same change occurs in just one layer. Indeed, the rate at which neighbors are reshuffled grows dramatically when the ordered clusters appear (see Sec. 2.1). The neighborhood composition changes significantly also between two blocks when the channels are downsampled.

Figure A.3: Overlap with layers 25, 88, 148 in ResNet152. Different background colors indicate different ResNet blocks

## A.4  Central kernel alignment vs overlap

Central kernel alignment (CKA) is the normalized squared Hilbert-Schmidt norm of the cross covariance operator between representations [2]. Like the neighborhood overlap it is invariant under orthogonal transformations and isotropic scaling but not to an arbitrary invertible linear transformation. This has been argued to be too a limitation for a similarity index between representations [2]. Gaussian CKA probes the local similarity between representations and can seen as a kernel smoothing of the neighborhood overlap presented in Sec. 2.1. In figure A.4-a we show the gaussian CKA (orange) and the overlap (green) with the output layer setting the kernel bandwidth to 0.2 times the average distance with the first nearest neighbor. Linear CKA is equivalent to a CCA between representations in which the canonical variates are weighted by the corresponding eigenvalues [2]. Linear CKA steadily increases already in the early layers of the network (see Fig. A.4-a blu profile).

Figure A.4-b shows how the gaussian CKA with the output is affected by different choices of the kernel bandwidth $\sigma$. The smaller is $\sigma$ the sharper is the transition measured by the index.

Figure A.4: (**a**:) Linear CKA (blu), overlap $\chi^{l,out}$ (green) and gaussian CKA (orange) with the output layer in ResNet152 for a subset of 5000 ILSVRC2012 images. We kept 50 classes and 100 images per class and set the kernel bandwidth $\sigma$ to 0.2 times the average distance with the first nearest neighbor $\overline{d_1}$ . (**b**:) Gaussian CKA with the output layer as a function of the kernel badwidth $\sigma$: $\sigma = 0.1\overline{d_1}$ (blu), $\sigma = 0.2\overline{d_1}$ (orange), $\sigma = 0.5\overline{d_1}$ (green), $\sigma = \overline{d_1}$ (red), $\sigma = 2\overline{d_1}$ (pink).

## A.5 Overlap and intrinsic dimension profiles in different datasets

In this section we compare the overlap with the ground truth labels $\chi^{l,gt}$ and the intrinsic dimension (ID) profiles of different dataset of incesing complexity in ResNet152.

The top panel of figure A.5 shows $\chi^{l,gt}$ for a ResNet152 architecture trained on MNIST [3] modMNIST, CIFAR-10 [4] and ImageNet [5]. To generate the modMNIST dataset we resize the dimension of the MNIST digits by a factor ranging from 0.2 to 0.4 and moved them in a random location of the image. We finally scale up the size of the images to 224x224 pixels. We trained MNIST and modMNIST for 10 and 20 epochs respectively using Adam optimizer [6] with default parameters (lr=0,001, betas=(0,9; 0,999)); we trained CIFAR10 for 120 epochs with stochastic gradient descent with momentum (lr = 0.1, momentum = 0.9), decreasing the learning rate by a factor 10 after 60 epochs; we used the PyTorch pre-trained ResNet152 model for ImageNet.

Figure A.5: Overlap with the ground truth labels (Top) and intrinsic dimension profiles (Bottom) in ResNet152 for different datasets: MNIST (red), CIFAR10 (orange), modMNIST (green), ImageNet (blue).

MNIST can be directly classified with high accuracy with a $k$-NN estimator. Consistently already in the input layer $\chi^{l,gt} \approx 0.78$ and reaches one from conv3 onwards. In modMNIST and CIFAR-10 the categories are only 10, therefore the initial values of $\chi^{l,gt}$ are larger, the lag phase is shorter the one of ImageNet. While qualitatively, $\chi^{l,gt}$ behaves similarly in modMNIST, CIFAR-10 and ILSVRC2012, the transition of $\chi^{l,gt}$ seems to be sharp only for the ILSVRC2012 dataset, and is therefore likely related to the complexity of the prediction task.

Bottom panel shows the intrinsic dimension (ID) for the same datasets across the checkpoints layers of ResNets152. The higher the complexity of the dataset the more are the factors of variations encoded in the embedding manifold, the higher is the ID. For complex datasets like ImageNet the ID has the hunchback shape reported in [7], while for MNIST and modMNIST it is almost constant, and it takes much smaller values, uncorrelated with $\chi^{l,gt}$. This supports the hypothesis that the transition observed in the value of the neighborhood overlap is not necessarily related with a sharp change of the intrinsic dimension of the representation.

## A.6 Details of density peaks appearing between layer 142 and 148

In figure A.6 we report a visualisation of the the density peaks appearing during the "nucleation transition" of Resnet 152. In particular, the image shows the size and approximate composition of the peaks present in the layers 142, 145, and 148. As discussed in Section 3.3, in layer 142 the data density is dominated by two large peaks composed of images of animals and artifacts respectively. This structure is visible in panel (a), in which one can easily identify the two large peaks. In the subsequent layers, the animal and artifact peaks break down into small peaks containing images of the same class. The process happens in a hierarchical fashion: peaks corresponding to multiple classes sharing a lot of semantic similarities appear first, and subsequently break down into smaller peaks corresponding to the single classes. This phenomenon can be observed in panels (b) and (c). For instance, in layer 145 (panel (b)) one can clearly identify peaks corresponding to certain kinds of arachnids (wolf spider, harvestman, tick, ...), insects (black and gold spider, leaf beetle, barn spiders) 4-wheel means of transportation (beach wagon, convertible, minivan), dogs (Samoyed, keeshond, chow), and so on. In layer 148 (panel (c)) this process continues and one finds many more peaks, corresponding either to single classes (e.g., iPod, piggy bank and beer bottle) or to groups of similar classes. At the end of the nucleation process described, from layer 152 (not shown here) one finds approximately one peak corresponding to each class label.

Figure A.6: Composition of density peaks in layers 142 (**a**), 145 (**b**) and 148 (**c**). The x-axis indicates the size of the peak, the y-axis reports the categories represented with more than 150 points in the peak. Consecutive dots ("...") indicate that more than three categories are well represented in the peak. The peaks are ordered from the smallest to the largest from top to bottom.