[Reviews · NeurIPS 2020]

Review 1

Summary and Contributions: The paper introduces an analysis on the probability densities for different layers of a Deep Convolutional Network. Particularly, authors focus on analysing a network trained in the ImageNet dataset, and through the paper describe the changes on the activation statistics when advancing to different layers.

Strengths: - Understanding how DCN is a relevant topic for the present and the future of the field. I think paper in this direction are very relevant for the community to understand and improve the models. - The paper does a very accurate description of the density shape for different levels of depth in the network. The paper is very well written and easy to follow. - The statistics presented by the authors to interpret the network activation are very intuitive and easy to follow by the reader. Paper figures describe the behaviour of the network very well. - The ImageNet dataset is one of the standard benchmarks in the field. Understanding the behaviour of the network in this dataset is relevant.

Weaknesses: - Authors miss to cite and comment some very related work in network interpretability such as: Bau et al, CVPR 2017. Network Dissection: Quantifying Interpretability of Deep Visual Representations. I think the work in Network Dissection is directly relevant to the topic discussed in the paper, particularly for its hierarchical analysis as well as focus on ImageNet. - It would also be interested to get analysis on multiple runs and other datasets. The paper is very interesting and has good insights, but the reader might wonder whether this effects are artifacts specific of some particular training run or this particular dataset. I think it would be good to have more variability to better understand the dynamics. - Why did authors restrict themselves to a smaller number of categories? I think it would be interesting to understand the behavior on a larger category set. - Have authors evaluated qualitatively some of the metrics? I think it would be interesting to see whether, additionally to the quantitative evaluation, the images have some features in common. After rebuttal: The authors have addressed most of my concerns and I will update my score to 7.

Correctness: Yes, the paper reads well and it is technically sound. The claims and the method seem correct.

Clarity: Yes, the paper is well-written and all the parts are well described.

Relation to Prior Work: As mentioned in the weaknesses section, prior work is not well described as they missed an important piece of work which is very related to the paper. They are missing: Bau et al, CVPR 2017. Network Dissection: Quantifying Interpretability of Deep Visual Representations.

Reproducibility: Yes

Additional Feedback:


Review 2

Summary and Contributions: This paper shows several analyses of internal representations of CNNs trained for ImageNet classification task. The main finding is that the representations evolve not smoothly but suddenly in the successive layers. It also presents how peaks of distributions of the representations emerge in the layers, from which a tree structure is obtained that show a hierarchical structure of object categories.

Strengths: - The results show intriguing natures of intermediate activations of deep CNNs. - The tools used in this study may be novel, i.e., computation of neighborhood overlap and finding peaks of the distribution of the layer representation.

Weaknesses: - The results do not provide us with significantly novel insight into the internal representation of CNNs. We are familiar with most of the observations in this paper, as similar attempts in previous studies. - It may be somewhat novel that the layer representations evolve rather quickly in several specific layers, not smoothly over all the layers, when the neighborhood overlap is employed to measure the speed of their evolution. However, if so, it is unclear how unusual it might be. If the representation evolves at a constant pace in the layers, others may say that it contradicts with our intuition or belief. - Moreover, it is not obvious in what way this finding is useful. - Although it is suggested that a more rational design of network architectures and/or learning methods will be possible, I don't understand how to do this in practice. After reading the author feedback, I understand the author's claim about the usefulness of the results they reported in this study. However, I'm still skeptical of the novelty. I do have similar concerns to R3. I think this is a neat study but is not above the bar of acceptance of NeurIPS.

Correctness: From what I understand, I think that the claim and method are correct.

Clarity: The paper is well written.

Relation to Prior Work: I think the paper properly cites the previous relevant studies, but there will be discussion on the novelty of the findings of this study.

Reproducibility: Yes

Additional Feedback:


Review 3

Summary and Contributions: The paper provides a novel approach to visualizing the feature representation in CNNs by applying an approach to analyzing the modes in high-dimensional feature data. Several novel claims are made regarding the representations learned by CNNs based on the representation analysis.

Strengths: The paper tackles an important problem as the nature of representation learning in deep models is a fundamental research question. The approach appears to be a novel application of existing data analysis methods.The four hypothesis listed in lines 65-77 are intriguing.

Weaknesses: The primary weaknesses are 1) lack of novelty, 2) concern as to whether the analysis method is advantageous and appropriate for understanding representation learning in CNNs, and 3) lack of convincing evidence that the four stated hypotheses are valid. - Novelty: The analysis method is a fairly straight-forward application of [14, 15] to the sets of activations produced by CNN layers. Moreover, as argued in A.4, the method is also closely-related to CKA [3]. Thus the novelty comes primarily from the specific hypotheses raised by the authors and the methods used to test them. Note that this is not a substantial weakness, in that if the findings were interesting and the evidence was persuasive then acceptance would be merited. - Neighborhood overlap: It is not clear what is the benefit of this approach vs CKA or other CCA methods. The need to pick a certain number of discrete neighbors seems disadvantageous. It is also unclear what transformations the comparison is invariant to. The paper could be strengthened by comparing the proposed method to CKA and related correlational analyses to highlight any similarities or differences. - Sec. 2.2: It is not sufficiently clear that the method from [14] is yielding valid results. In general, the structure of the high-dimensional pdf may be complex, and identifying peaks and saddle points may be difficult. The description of the method is not sufficiently clear to understand what assumptions it is making. Examining [14], it contains statements (Sec. 2, 1st sentence): "As in standard Density-Peaks clustering we here assume that the density peaks are surrounded by neighbors with lower local density and are at a relatively large distance from any points with a higher local density." How can we tell if this assumption is valid? How will the method perform for complex densities that lack well-separated peaks? - Hypothesis 1: The findings for the neighborhood structure and its correlation with the class conditional distribution (blue and orange curves) is not surprising, it just shows that later features have learned to separate the classes. The claim for nucleation hinges on the green curve, and the evidence is not compelling. It appears that for layers 1-142 there is no nucleation. It is confusing that the x axis of this graph does not follow a constant scale (gap from 34-142 is the same distance as 1-10). Any "nucleation" seems to be happening only for the last few layers (142-153). It is not clear that this means anything, it might simply be a consequence of learning more complex features, for example. The paper could be strengthened by framing the hypothesis in terms of a quantitative test that could be evaluated using data. - Hypothesis 2: The claim that the early distributions of activations are significantly different from pixel distributions is supported by the first datapoint in the green curve of Fig. 2, but it is not surprising. The discussion in Sec. 3.2, which seems to be supporting the hypothesis that there is "a single probability peak," is unclear. What is the source of the hub images which are illustrated? How do the graphs in Fig. 3 support this hypothesis? The claim that the activations follow unimodal distributions seems to be in conflict with previous visualization results for deep models, which show layers responding preferentially to a diverse (multimodal) set of "mid-level" image properties. The paper could be strengthened by providing a quantitative test for the hypothesis. - Hypothesis 3: It is difficult to draw any strong conclusions from the 2D visualization in Fig. 4(a). If some quantitative hypothesis test were performed, this figure could be useful in supporting the analysis. But there are many ways that low dimensional embeddings of high dimensional structures can be misleading, and it shouldn't be the primary source of evidence for a claim. It is not clear what the ARI analysis illustrated in Fig. 4(b) is based on or how it is performed, and it seems to be subject to the difficulties in correctly identifying peaks referred to earlier. - Hypothesis 4: This claim makes sense in terms of the feature transformations mapping the data into separable distributions. The visualization in Fig. 4(c) is interesting and potentially useful. It is more believable that peaks can be easily detected in the densities arising in the final layers.

Correctness: See the discussion of weaknesses above for concerns regarding the methodology and claimed hypotheses.

Clarity: The paper is well-written overall. But the discussion of the methods could be improved to make the paper more self-contained and make it clear what assumptions the approach is making and what issues may arise in its application to the problem. Currently it would not be possible to implement the method based on the description in the paper without reading the prior methods in detail.

Relation to Prior Work: The paper could be strengthened by connecting the four hypotheses more clearly to prior work on representation learning. To what extent are they supported by or contradicted by the existing literature? More discussion of this point would strengthen the paper.

Reproducibility: No

Additional Feedback: There were a couple of small typos: 125: if [there] exists a point 249: assess ** Comments following Rebuttal ** The rebuttal was valuable and provided additional evidence for the method and effectively addressed some of my concerns. I particularly find the new results compelling in providing converging evidence for the claims in the paper. While I have some remaining concerns about the approach, I am persuaded that this paper contributes a novel tool for understanding DCN, and makes a sufficiently useful contribution (provided that the new results and arguments from the rebuttal are incorporated into the final manuscript). Based on this, I am upgrading my score to the accept side of the column.


Review 4

Summary and Contributions: This paper studies how the data representation within the discriminative deep neural networks evolves with respect to the depth. The authors show that the data representation will concentrate together and formed individual density peaks as it goes deeper. The density peaks aligned with the gt labels appear very late (close to the output) in the network and appear all of a sudden. The structure of the peaks also have some connections to the semantic relationships of the categories.

Strengths: - The approach shed light on how the data representations evolved within the deep neural networks: the evolution process is not smooth but abrupt. - the presented approach seems to be robust to different hyper-parameters. With different size of neighbor k, and different amount of data N, the results are all more or less the same.

Weaknesses: - more analysis on other network architectures are required. analyzing merely the ResNet and VGG family is not enough. the authors shall include results on other families such as DenseNet, GoogleLeNet, etc. - how do the authors estimate the local volume density with kNN (L116)? if one simply uses the region that contains k nearest neighbors to define density, it will have some issues. For instance, it will have a lot of discontinuities. Can the authors be more specific on this and provide more explanations? - I'm slightly confused about Fig. 2(a). the number of measurements for \chi^{l, l+1} is different to (more than) those of \chi^{gt} and \chi^{out}. what happened? there seems to be more variantions for \chi^{l, l+1} when the measurement scale is more fine-grained - does it apply to \chi^{gt} and \chi^{out} too? - the authors have argued that the observation may open the opportunities to several interesting directions, such as helping us design better architecture, enabling more powerful training scheme, which I totally agree. But it would be better if the authors can actually show some preliminary results on some of these directions. This will make the submission way more solid. Currently I feel like the depth of the paper can be improved a bit. The observation is cool, but then what?

Correctness: -

Clarity: - how is Fig. 2(b) computed? my hypothesis is that it's a historgram over all data instances? ie without the outter summation and division in Eq. (1)? please confirm and make it more explicit (L1651-166). - The paper consists of quite a few super long sentences, which increases the difficulty of reading. Many sentences can be written in a more concise fashion..

Relation to Prior Work: -

Reproducibility: Yes

Additional Feedback:

[Author Response · NeurIPS 2020]

We thank all the referees for their helpful comments and the constructive feedbacks. All the referees agree that the paper is a relevant and meaningful contribution. Even the most negative referee (**R3**) recognizes that the results we obtain are "intriguing". **Our main result is the** observation that the hidden representations become ordered via a **sharp transition near the end of the network** which is sharper for deeper networks and for more complex datasets. **R2** recognizes the novelty of this result, but

then s/he claims that *"it is unclear how unusual it might be"*. This is presented as a weakness, but if we are the first to make such an observation and present quantitative evidence supporting it, we believe this should be considered a strong point, even if the results are compliant with the intuition of the referee. Importantly, to address the concerns raised by **R2** and **R4** about the usefulness of our results, **we point out at least three possible practical applications of our findings**. **1)** The analysis of the density peaks hierarchy can allow an optimal truncation of the network in transfer learning schemes, exploiting the semantic hierarchy. **2)** The profiles of $\chi^{gt}$ can be used to check if the architecture is oversized in relation to the complexity of the classification task. **3)** Our observation on the existence of a sharp transition and our tools to characterise it could help in designing better performing architectures and training schemes. For instance the transition could be facilitated by "seeding" it using a metric loss function similar to that used for Siamese and Triplet networks [SIMBAD, V 9370, pp 84-92, 2015]. We will clarify these (overlooked) implications in the revised manuscript.

Following **R1** and **R4** we corroborated our results by performing new tests on GoogLeLeNet and DenseNet121 pre-trained on Imagenet (Fig. a). We also trained a small convnet on the UrbanSound8K dataset reaching a $88\%$ test accuracy (Fig. b). In both cases we found the same characteristic transition curve shown in the manuscript (Fig. 2c). **R1** suggests that more categories could be analysed to improve the robustness of our results. This would be impractical with current HPC infrastructure. However, as also acknowledged by **R4**, the scaling analysis done in A1 makes us very confident that the results would not change significantly using more points. **R1** and **R3** mentioned the possibility of linking more strongly to existing literature in the field. Thanks to the useful direction indicated by **R1**, we found two very revant works: the network dissection analysis of [CVPR, pp. 3319-3327, 2017] and the linear probe analysis of [arXiv:1610.01644, 2016]. These works tackle the challenging problem of understanding the hidden representations of deep CNNs, but our analysis is complementary to theirs because **1)** the analysis of data probability densities has never been performed and, as a consequence, **2)** the results we obtain (lines 65-77 of the manuscript) have not been previously reported. A critical discussion of our work in comparison to the above references will be added to the manuscript.

**R3** is particularly critical of our submission. Her/his most important concern is that *"it is not sufficiently clear that the method from [14] is yielding valid results"*. It is important to stress that the main assumption of the Density Peak clustering [Science, V. 344, p. 1492, 2014] (i.e., that *"the density peaks are surrounded by neighbors with lower local density"*) has been verified by several groups independently, and that the method has been tested and used extensively receiving thousands citations. One of the goals of our work is demonstrating that this approach is useful also for analyzing the activations in a DNN. In addition, as indicated by **R1**, **R2** and **R4** our work is fully reproducible since: **1)** we provide a self-contained and easy-to-run jupyter notebook in the SM for generating the main results of the paper and **2)** a detailed explanation of the method can be found in the literature.

In connection to Fig. 4a **R3** also remarks that *"low dimensional embeddings of high dimensional structures can be misleading"*. We stress here that for our analysis **we *do not* require any low dimensional embedding of the data**, and this is one of the greatest advantages of the method we use. In other words, Fig. 4a is *not* used as source of evidence for our claims but only to aid the visualisation of the results which are obtained independently through the analysis of the density peaks. **R3** raises a concern about the invariance properties of the overlap $\chi$. Equation (1) shows that $\chi$ is computed from the product of two adjacency matrices which are built using the euclidean distances between images, as such they are invariant to orthogonal transformations but not to any arbitrary linear transformation of the activations. A third concern of **R3** is that *"Any nucleation seems to be happening only for the last few layers (142-153)[...] it might simply be a consequence of learning more complex features"*. We explicitly state in the manuscript that the transition from disordered to ordered neighbourhoods is a consequence of progressive learning. However, what we call "nucleation" **does not happen progressively in all last layers, but as a sudden change** immediately after layer 142. This, in our opinion, is a highly non-trivial result.

**R3**'s concern on Hypothesis 4 seems more like a positive remark than the description of a weakness.

Regarding the question of **R3** and **R4** about Fig. 2a, we chose to display evenly spaced the pooling layers and the outputs of the four ResNet blocks, since these are "architectural milestones" where the networks downsample the images. On the contrary we computed $\chi^{l,l+1}$ between each couple of layers, this is why the number of measurements is different in Fig. 2a. **R4** Fig. 2b is indeed the histogram over all the data instances without the outer summation and division of Eq. (1). We will clarify this in the final version of the manuscript.

[Meta-Review · NeurIPS 2020]

All reviewers found the work compelling, and the analysis of probability densities and presented hypotheses broadly interesting to the community aiming to understand and visualize the internals of neural networks. In discussion, reviewers found the rebuttal convincing and the additional experiments beneficial to strengthening the hypotheses in the paper. For the camera-ready version, please include the additional figure and address any remaining minor concerns.